# Testing Seefeldt’s Proficiency Barrier: A Longitudinal Study

**DOI:** 10.3390/ijerph19127184

**Published:** 2022-06-11

**Authors:** Fernando Garbeloto dos Santos, Matheus Maia Pacheco, David Stodden, Go Tani, José António Ribeiro Maia

**Affiliations:** 1CIFI2D, Faculty of Sport, University of Porto, R. Dr. Plácido da Costa 91, 4099-002 Porto, Portugal; matheus.lacom@gmail.com (M.M.P.); jmaia@fade.up.pt (J.A.R.M.); 2Department of Physical Education & Athletic Training, University of South Carolina, Columbia, SC 29208, USA; stodden@mailbox.sc.edu; 3School of Physical Education and Sport, University of São Paulo, Av. Professor Mello Moraes, 65, São Paulo 05508-030, Brazil; gotani@usp.br

**Keywords:** intervention program, motor development, fundamental movement skills, transitional motor skills, specific sport skills

## Abstract

The idea that proficiency in the fundamental movement skills (FMS) is necessary for the development of more complex motor skills (i.e., the proficiency barrier) and to promote health-enhancing physical activity and health-related physical fitness levels is widespread in the literature of motor development. Nonetheless, to the best of our knowledge, there is no study assessing whether children presenting proficiency below a specified proficiency barrier would demonstrate difficulty in improving performance in more complex skills—even when subjected to a period of practice in these complex skills. The present study tested this. Eighty-five normal children (44 boys) aged 7 to 10 years participated in the study. The intervention took place during 10 consecutive classes, once a week, lasting 40 min each. Six FMS (running, hopping, leaping, kicking, catching and stationary dribbling) and one transitional motor skill (TMS) (speed dribbling skill) were assessed. The results showed that only those who showed sufficient proficiency in running and stationary dribbling before the intervention were able to show high performance values in the TMS after intervention. In addition, in line with recent propositions, the results show that the basis for development of the TMS was specific critical components of the FMS and that the barrier can be captured through a logistic function. These results corroborate the proficiency barrier hypothesis and highlight that mastering the critical components of the FMS is a necessary condition for motor development.

## 1. Introduction

Motor development models posit a hierarchical structure in the development of movement skills [1,2,3] based on the fundamental idea that previous experiences influence, if not determine, what can be performed and learned in the next stages of life. Within this hierarchy, Seefeldt [4] proposed that children who do not learn to perform fundamental movement skills (FMS) proficiently (a set of motor activities that are believed to be the basis for many activities; e.g., running, stationary dribbling, kicking) will have difficulty in learning more specific skills (FMS combination—e.g., straight speed dribbling; culturally or sport-specific skills—e.g., kicking pattern in soccer; kicking pattern in American football). This condition is known as the proficiency barrier.

The proficiency barrier has been noted as an important phenomenon to be investigated in the field of motor development [5]. This is due to emerging evidence suggesting that adequate levels of FMS would positively associate with participation in a broad array of physical activities [6] and, consequently, promote health-enhancing physical activity and health-related physical fitness levels [7,8,9]. Due to the generally consistent positive associations between FMS levels, PA and various measures of health-related physical fitness, researchers conducted examinations of whether a proficiency barrier may exist relating the impact of FMS proficiency to physical activity levels [10] and health-related physical fitness [9,11].

Despite the importance of the posited relation between motor development and health-related physical activity habits, the proficiency barrier directly refers to the relation between FMS and transitional movement skills (TMS)—a set of motor activities that would be in-between FMS and sport-specific skills in terms of collective movement pattern integration [4,12]. According to Seefeldt’s proposition, there may be a level of performance that would be necessary for the acquisition of more complex skills. To the best of our knowledge, there are only two studies that have tested the proficiency barrier proposition empirically, addressing the relation between FMS and TMS considering Seefeldt’s descriptions [13,14]. 

Costa et al. [13] longitudinally evaluated the relationship between volleying and throwing (FMS) on the volley serve (TMS) through different practice schedules (constant-random and random) in the TMS. Their findings support the proficiency barrier in showing an improvement in the movement pattern of the TMS for those with higher levels of FMS. However, Costa et al. [13] is limited in providing determining evidence as skill levels (proficiency/non-proficient) were defined arbitrarily (i.e., only those with maximum criteria in all FMS components were considered proficient). While a specific level of skill was not assumed by Seefeldt [4], Pacheco et al. [14] (see below) demonstrated that such high level of proficiency is not necessary. A second issue is that when a direct relation between FMS and TMS was assessed, the authors did not assess the barrier per se; rather, their statistical analyses addressed the relationship from a linear perspective.

The second study, conducted by Pacheco et al. [14], also showed, as predicted, that children with better performances in the FMS (e.g., running and stationary dribbling) were the ones who performed better in the straight speed dribbling (i.e., running and dribbling the basketball with their preferred hand) that integrated associated FMS. Different from Costa et al.’s [13] results, Pacheco et al. [14] revealed that such relationship follows a logistic function—a non-linear function that would capture the proficiency barrier as would be expected from the dichotomous language separation of a true “barrier”. Additionally, the study was able to identify the presence of critical antecedents, a core assumption in the original formulation [4]. That is, only the children who already learned specific movement components from two FMS (running and stationary dribbling) were able to demonstrate high proficiency in the studied TMS (straight speed dribbling).

Notwithstanding Pacheco et al.’s [14] findings and putative practical implications for learning skills, their investigation did not test if performing above the barrier is a necessary condition to more effectively improve performance and learn the TMS from a longitudinal or experimental design perspective. In other words, to what extent do children below a specified proficiency barrier, even when subjected to a period of practice, improve their TMS performance and overcome the barrier? (In fact, Seefeldt [4] stated that “children deprived of learning the fundamental skills have difficulty when they attempt to learn the transitional motor skills” (p. 316). This could lead to several observations: for instance, lower learning rate, a diminished ceiling performance for individuals with low FMS proficiency, and the requirement for differentiated practice. Provided this underdetermination of the hypothesis, in light of the nonlinearity demonstrated in Pacheco et al. [14], and the lack of effect of different practice schedules for low competence in Costa et al. [13], we hold to a “strict” proficiency barrier hypothesis where the barrier (low FMS proficiency) limits learning of the TMS).

Thus, there is no convincing evidence supporting the proficiency barrier within the aforementioned line of reasoning. Therefore, the aim of the present study is to empirically test the proficiency barrier proposition—that is, if specific motor experiences at a given period can influence, or are a necessary condition, for later motor skills’ development. Following the same procedure outlined in Pacheco et al. [14], we first test the existence of a proficiency barrier by observing the existence of a threshold relating to the possibility of improvement in TMS performance after an intervention program. Second, we verify the existence of critical antecedents by relating proficiency in given movement components of the FMS to TMS proficiency after an intervention program. 

## 2. Materials and Methods

### 2.1. Sample

The data for the present paper are part of a longitudinal research project that investigated the effects of different intervention programs on FMS, TMS and sport-specific skills performance over a 2.5-year period [15]. One of the main goals of this project was to compare the effect of specific (practice of the FMS) and general sessions (traditional PE classes) on changes in FMS proficiency levels. 

All students of a private school located in São Paulo (Brazil) were enrolled in grades 2 to 5 of primary school (from 6 to 10 years of age), and while taking part in mandatory physical education classes (twice a week), received the consent form and were invited to participate. Only those with informed consent, dated and signed by their parents/legal guardians, were considered as participants. The exclusion criterion was to present any physical and/or intellectual disability that could impair their response in the motor assessments. In total, 85 children (44 boys) aged 7 (*n* = 25), 8 (*n* = 9), 9 (*n* = 39) and 10 years (*n* = 12) (mean age 8.39; SD 1.09) participated in the study (~30% of the total number of children enrolled between the 2nd and 5th grade of the school where the research took place). Part of the sample (~50%) practiced sports after the school period: 28% indoor soccer, 13% judo, 9% swimming, 2% volleyball and 1% swimming. The sample for the present study was the same used in Pacheco et al.’s [14] report. Their data, nonetheless, refer to the results of the baseline testing (i.e., at the entry of the study). 

### 2.2. Intervention

The intervention took place between March and May of 2016, with 10 consecutive teaching classes, once a week, lasting 40 min each during one of the two weekly Physical Education classes included in the school curriculum. In the other class of the week, the physical education teacher continued with the school-scheduled content (without any intervention from the research team). Only one physical education teacher who was responsible for students from the 2nd to the 5th year was also responsible for application of the intervention program. Before starting the intervention program, the Physical Education teacher met with one of the researchers to receive instructions about the intervention program (e.g., how to provide movement instructions and the structure of each class) and to assist in the preparation of the content of the 10 classes. Because the intervention was carried out during physical education classes, even those students who did not bring the signed consent form participated in the intervention program, although they did not have their performance assessed. Further, the same intervention was implemented to all students, regardless of their age. Only the materials used (e.g., the type and ball size) and the goal of the task (e.g., the height of the basket) were changed according to the needs of each age group.

All classes consisted of a brief initial warm-up period (~3 min), followed by games and exercise activities that encouraged children to practice two FMS—stationary dribbling and running (without driving an object)—and one TMS—speed dribbling (e.g., run while dribbling a basketball) in different situations (e.g., with and without an opponent, with changes in direction, with different speeds). According to Tani [16], practicing the same skill in different situations is a necessary condition to learning new motor skills [16,17]. A complete description of the intervention program is available in Appendix A.

### 2.3. Procedure

Fundamental movement skills: six FMS (running, hopping, leaping, kicking, catching and stationary dribbling) were assessed with the TGMD-2 test battery [18]. In brief, this test battery comprises 12 fundamental movement skills (run, gallop, hop, leap, horizontal jump, slide, overhand throw, underhand roll, stationary dribble, striking a stationary ball, catch and kick) validated for children from 3 to 10 years of age. Each skill has 3 to 5 criteria scores that qualitatively assesses the movement pattern, and the scoring is based on whether each criterion was present (1) or not (0). 

The test application followed the procedures outlined by Ulrich [18]. The sum of the scores of both trials was used as a performance measure so that each child could reach a maximum of 46 points in the sum of the six FMS and 16 points in the sum of running and stationary dribbling.

Transitional movement skill: The TMS was assessed using a checklist that assesses proficient performance on a speed dribbling skill which has been shown to be valid and reliable [19]. The speed dribbling test required participants to cover, as fast as possible, 18 m in a straight line by running and dribbling the basketball with their preferred hand. The checklist has a total of nine qualitative binary criteria similar to the TMGD-2 scoring procedures (observance of criterion 1 score, otherwise 0). We chose this TMS because it combines running and stationary dribbling, i.e., two FMS, and the assessment protocol is similar to TGMD-2 guidelines (based on movement-component criteria).

FMS and TMS were recorded with a camera Sony HDR-PJ540 (60 Hz). The test application followed the procedures outlined in Santos et al. (2020). Only one team member (the first author) rated all 85 children’s videos. A week later, videos of 20 children were rated again (intra-rater agreement) in a random order with its agreement calculated with Cohen’s κ. For the FMS, assessed by TGMD-2, Cohen’s κ ranged from 0.75 in leap to 0.95 in stationary dribbling. In speed dribbling, the intra-rater agreement was calculated for each component (nine in total), and Cohen’s κ ranged from 0.73 (component 9) to 0.91 (component 6). Inter-rater agreement using three different raters assessed 50 children (based on different levels of performance). The average Cohen’s κ was high (κ = 0.77).

### 2.4. Data Analysis

The first analysis was to test whether each of the seven tested skills (six FMS and one TMS) improved during the period of intervention. Provided the data failed to follow parametric assumptions, each skill was tested separately through a bootstrap *t*-test using the Resampling Statistical Toolkit [20] with 2000 iterations.

The second analysis tested the “cross-sectional” proficiency barrier. This analysis provides means to replicate the results of Pacheco et al. [14] and it is a necessary condition for further analyses. We compared a linear fitting (function with no proficiency barrier) against the logistic function (function with proficiency barrier) relating the sum of criteria in the two FMS of interest (running and stationary dribbling) and the sum of criteria in the TMS (speed dribbling). In the logistic function,
TMS = *α* + (*β* − *α*)/(1 + *exp*(−(FMS − FMS_b_)/*δ*)))(1)

*α* is the TMS average performance below the barrier, *β* is the TMS average performance after the barrier, FMS_b_ is the value at which the function achieves 50% of change (the “FMS barrier value”), and *δ* is the modifier of the slope of the function. All these are fitted parameters from data. The logistic function encompasses the barrier “idea” by separating high and low levels of TMS through a threshold value as a function of the FMS. Thus, those above a given FMS number of criteria (barrier) could demonstrate high TMS proficiency while those below the barrier could not. To make sure that both logistic and linear functions had the same number of fitted parameters, we constrained both FMS_b_ and *δ* with the same values (FMS_b_ = 12.48; *δ* = 0.7586) as in the previous study [14]. The comparison between these two models was based on the adjusted *R*^2^. 

Still on the cross-sectional proficiency barrier, we fitted—with a bootstrap procedure—the logistic function with no constraints as to observe whether the fitted parameters (mean and confidence intervals) encompassed the values found in the previous study. The bootstrap procedure was performed with 2000 iterations.

The third analysis tested the longitudinal proficiency barrier. That is, we wanted to investigate whether only those with FMS (sum of criteria of running and stationary dribbling) above the barrier at the baseline would demonstrate improvements in TMS. To fully understand such an issue, one needs to test the relationship between FMS performance in the first data collection with the change in TMS from the first to the second data collection, controlling for the improvement in FMS from the first to the second data collection and TMS performance in the first data collection. The strategy of fitting a logistic function—performed in the cross-sectional analysis described above—is not suitable here as the number of fitted variables becomes too large for the number of individuals (degrees of freedom). For this reason, we used a simpler strategy that suffices for the current aim. We performed a bootstrapped *χ*^2^ test for the association of individuals above/below the FMS barrier with individuals above/below the TMS maximum below the FMS barrier. Using the values observed in the previous study, we categorized individuals in terms of showing more than 11 (out of 16) criteria in the FMS and 9 (out of 16) criteria in the TMS. These values were selected provided those showing less than or equal 11 criteria in the FMS showed only less than or equal to 9 criteria in the TMS in Pacheco et al. [14]. Two *χ*^2^ associations were performed: between the FMS categories from the first data collection and the TMS in the second data collection, and (for control) between the FMS categories from the second data collection and the TMS in the second data collection. A necessary finding to consider the proficiency barrier (in both associations made) was that there were no cases in which low FMS were related to high TMS. Any other case was theoretically valid.

Finally, the fourth analysis tested whether the critical antecedents (observed at the first data collection) conformed to a logistic relation with the TMS of the second data collection. To find the critical antecedents, we considered the covariance pattern that described the set of TGMD criteria and demonstrated the proficiency barrier relationship with TMS in the first data collection. For this, we considered as proficient in a given criterion only those who scored in the criterion in both trials (in the first data collection). Then, we performed Horn’s Parallel Analysis with tetrachoric correlations testing for “factors” (covariance patterns) that significantly explained the data when compared to shuffled data using the pa_rule_polychoric_missing function [14,21]. After finding the critical antecedents, we tested whether there was a relationship between the critical antecedents and the sum of criteria of the TMS in the second data collection.

All analyses were performed in MATLAB R2020b Update 1 (9.9.0.1495850) (MathWorks^®^) and considered significant at *p* < 0.050. Further, the full MATLAB Live Scripts are presented in Appendix A.

## 3. Results

### 3.1. Improvement in Performance Given the Intervention

Table 1 shows the descriptive data and the results of the bootstrap *t*-test performed for all six FMS and one TMS tested in the present study. As noted in Table 1, running (*t* [84] = 3.63; *p* < 0.001), stationary dribbling (*t* [84] = 4.48; *p* < 0.001), kicking (*t* [84] = 1.78; *p* = 0.007), and speed dribbling (*t* [84] = 4.69; *p* < 0.001) showed improvements from the first to the second data collection. All other skills did not show any significant improvement.

### 3.2. Cross-Sectional Proficiency Barrier

Figure 1 shows the adjusted linear (Figure 1a) and logistic functions (Figure 1b) to the relationship between FMS in the second data collection to the TMS in the second data collection. The adjusted R^2^ values for the linear and logistic relations were 0.26 and 0.27, respectively.

When we performed the bootstrap on the logistic function—freeing FMS_b_ and *δ*—the mean and the 95% confidence intervals for FMS_b_ and *δ* were, respectively, 12.24 (CI_95%_ = [10.56, 15.35]) and 1.22 (CI_95%_ = [0.03, 3.50]). The values obtained in Pacheco et al. [14] (12.48, 0.76) are similar to the observed ones and clearly encompassed by the confidence intervals. Thus, it seems that the same “phenomenon” is observed in the first and second data collection.

### 3.3. Longitudinal Proficiency Barrier

Table 2 shows the relation between FMS categories (in the first and second data collection, respectively) with the TMS categories shown in the second data collection. The bootstrap *χ*^2^ tests showed a *χ*^2^ = 22.26 (CI_95%_ = [10.75, 35.51]) for FMS categories in the first data collection association with TMS categories and of 5.89 (CI_95%_ = [1.52, 13.50]) for FMS categories in the second data collection association with TMS categories. Given the critical value for *p* = 0.050 is *χ*^2^ = 3.84 and for *p* = 0.001 is *χ*^2^ = 10.83, the association for the FMS of first data collection is significant while the association for the FMS of the second data collection is not. However, the first requirement for a barrier is observed: there is no individual included in the low FMS category and high TMS results.

### 3.4. Testing the Longitudinal Barrier through the Critical Antecedents

After performing Horn’s parallel analysis, we found a single factor that included covariance in six out of eight criteria of both running and stationary dribbling (considering the first data collection). Figure 2 shows the relation between this factor and the TMS criteria observed in the second data collection. As observed, the logistic relation is well-observed with an *R*^2^ of 0.45. That is, only those who acquired the critical antecedents in running and stationary dribbling before the intervention were also able to show high performance values in the TMS (performance above the barrier) after the intervention program.

## 4. Discussion

The proficiency barrier is a well-known hypothesis derived from the hierarchy principle in motor development. Such principle is, explicitly or implicitly, postulated in many models of motor development [2,4] and proposes that specific experiences at a given age have profound influence (or are necessary) for later motor skill development. Despite the importance of the principle for a theory and, therefore, the hypothesis, few studies have addressed this important hypothesis. This was the aim of the present study.

Although interest in the proficiency barrier has grown in recent years [5,9,10], to the best of our knowledge, only two recent studies have been conducted to support its existence [13,14]. In addition to the linear relation found between good levels of performance in FMS and in TMS/sport skills [13], Pacheco et al. [14] provided three necessary conditions required to identify the proficiency barrier: (a) a non-linear relation (the logistic function) between FMS and TMS proficiencies; (b) a critical point of FMS performance that categorically separates those who are above and below the proficiency barrier (i.e., do or do not demonstrate high proficiency levels in the TMS); (c) a set of specific movement components that are the basis of the relationship between FMS and TMS (i.e., the critical antecedents). 

Despite their importance, the cited studies failed to demonstrate whether, even after a period of practice, individuals with fundamental movement patterns performance below the proficiency barrier would be unable to improve their performance. In this way, the present study asked children to participate in an intervention program where they would practice two FMS (running and stationary dribbling) and one TMS (speed dribbling), controlling their initial FMS and TMS status and tracking their improvements in TMS performance. 

The results showed that the intervention program promoted significant improvements in running, stationary dribbling and speed dribbling skill. In general, intervention programs with longer [22] or shorter durations [23], with similar [24] or different age samples [25], showed improvements in the FMS performance level [26,27]. Although the improvement magnitude for the participants in this study was around one criterion level, this represents a gain of 12.5% of everything that can be shown. Still, assessing advances in skill level based on TGMD criteria may be limited based either on a ceiling effect or in terms of how skills are assessed (i.e., limited discriminatory capability—presence or absence of a specific criterion). In the first data collection, 84% and 33% of children, in running and stationary dribbling, respectively, already demonstrated all possible eight components for each skill. Thus, for a large part of the sample, the assessment may not be appropriately sensitive to actual changes in skill that were promoted by the intervention and may partially explain the low magnitude of change for both skills. Third, another confounding factor could be the application of similar intervention for individuals of different ages. That is, the intervention could have been suboptimal for some individuals. Nonetheless, how much improvement is required to be considered sufficient is a debate that goes beyond the present study.

It should be noted that despite the low overall magnitude of improvement, only the skills that were directed by the intervention showed improvements. The lack of improvement in the other three FMS assessed (hopping, leaping and catching) allows us to suggest that the systematic practice during the 10 sessions in the intervention program was necessary to improve the stationary dribbling, run and speed dribbling. This result is in consonance with the statement that the FMS must be systematically practiced in order for improvement to occur; FMS proficiency does not happen “naturally” [11,26]. The only exception to the rule was kicking, which improved despite not being targeted by our intervention. Kicking improvement may have occurred, nonetheless, precisely because of systematized practice. In addition to Physical Education classes, 50% of the participants practiced a sport outside school hours, with soccer being the most practiced sport.

With the improvement in performance, 8 of the 11 participants who performed below the proficiency barrier in the first data collection exceeded the proficiency barrier after the intervention program. Despite overall performance improvement, the logistic function remained the best model describing the relationship between FMS and TMS proficiencies. The cross-sectional results remained independent of fixing parameters and fitting to the data or comparing the previous barrier values to fitted parameters, which supports the existence of the proficiency barrier. In other terms, the cross-sectional proficiency barrier was supported independent of the statistical test employed. In testing the longitudinal relations, we also found supporting evidence for the barrier. First, by categorizing participants in terms of their proficiency levels in FMS and TMS in both data collections, we found no violations of the barrier (i.e., no individuals with FMS performance below the barrier demonstrated good TMS performance). Second, the relations show that those with poor FMS scores in the first data collection fail to show high TMS proficiency in the second data collection. Thus, our results are the first longitudinal evidence of the proficiency barrier. These results also suggest a threshold value of the barrier for the relationship between running and stationary dribbling to speed dribbling (see the discussion below on critical antecedents).

It is important to point out that the logistic function seemed to show a slight improvement in variance accounted for (*R*^2^) when compared to a model not including a proficiency barrier (linear function) in the cross-sectional analysis. One could argue that such small increase in variance explained (0.01%) in the data is not enough to support the hypothesis—making a simpler phenomenon more complex with no great gain in explanatory power (challenging Occam’s Razor). However, the linearity observed seems to result from the fact that, in the second data collection, few individuals were below the barrier which constrains the data to the portion of the increase in TMS as a function of FMS. If we are to also consider the data presented in Pacheco et al. [14], we see that this apparent linearity disappears when a whole range of behaviors are considered. A similar issue also occurs for the results of the longitudinal barrier considering the FMS and TMS categories (both in the second data collection). Given the lack of individuals demonstrating low FMS in the second data collection, one would not expect associations to occur.

An interesting result is that those participants who exceeded the proficiency barrier in FMS from the first to the second data collection did improve their performance in TMS but failed to surpass our criteria of above-TMS performance in speed dribbling (Table 2). Tani et al. [17] suggest that good levels of performance in FMS are a necessary condition for the child to be able to modulate parameters (e.g., force, speed, relative timing and directionality) without losing proficiency in the skill. It can be speculated—in line with Tani et al.’s argument—that for FMS to be the basis for TMS, one must be able, first, to accommodate such variations in FMS and, only then, be able to appropriately apply those accommodations to improve a more complex skill. Thus, despite increases in the capacity to demonstrate the FMS, one must also be able to perform them flexibly and adaptively. This speculation favors a view in that not only consistency in movement patterns is required, but also flexibility.

An essential constituent of the proficiency barrier proposed by Seefeldt refers to the critical antecedents. The critical antecedents would be movement components that would be the actual required basis for TMS acquisition [4]. That is, they would represent the actual movement aspects that are required for one to learn the new skill. In the relationship between FMS and TMS investigated, we identified a single factor that included covariance in six out of eight criteria of both running and stationary dribbling (see below). The fact that these critical antecedents—extracted from the first data collection—demonstrated the similar logistic function when related to the TMS proficiency in the second data collection corroborates the results found by Pacheco et al. [14] and further supports the proficiency barrier. 

As discussed in Pacheco et al. [14], the critical antecedents observed in the present study refer to ball control in stationary dribbling and coordination between arms and legs in running. Clearly, speed dribbling requires the maintenance of ball control while a new coordination between legs and arms are present. This is similar to what was observed in the results of O’Keefe et al. [28] (but see below). They observed an interesting transfer effect between learning the overhead throw to javelin throw and badminton overhead clear. All movements investigated demonstrate a proximal-to-distal “whip-like” pattern when performed at an advanced level. In other terms, acquiring the capability to effectively generate and transfer energy through the kinetic-chain system via translation of the center of mass and increasing rotational torque across multiple joints while exploiting inertial characteristics of segmental masses facilitates transfer to different sport-specific skills.

In fact, more than the actual FMS proficiency, the critical antecedents seem to represent the core foundation of a potential specific theoretical development of the proficiency barrier and, more generally, to the principle of hierarchy. In Pacheco et al. [14], the authors postulated that the critical antecedents would be dynamical resources that would need to be learned before new movement patterns could be stably assembled. This would follow similar findings that Thelen and colleagues (see [29], for a review) found in infants before they demonstrated the walking pattern. The idea is that the critical antecedents are learned forms of interaction between organism and environment towards a goal (learning to interact with constraints [30]). This proposition seems to directly match the ideas proposed in the dynamical systems approach to motor behavior since these dynamical resources would be able to be generalized to new situations if similarities occur between task (and other) constraints (see [31,32]). Truly, there is a great separation between “qualitative” descriptions of dynamical resources and a formal theory relating them to facilitation in new tasks, but we believe this is the direction that should be followed.

We speculate that the critical components might also be the basis to discuss two other important aspects: the relation between motor development and physical activity [11] and the possibility to find the proficiency barrier assessing the FMS with other motor skills batteries. If Seefeldt’s [4] rationale is correct, these antecedents would be the basis for improvement in more complex skills. As such, this would potentially increase children’s participation in different sports and/or physical activities, thus reducing sedentary behaviors [7], and consequently, promoting health-enhancing physical activity and health-related physical fitness levels [8,9]. Interestingly, the Get Skilled Get Active test [33] points to movement components that learners are most capable of demonstrating, considering their developmental status. It might be questioned whether these components—which are presented in terms of the order that they are mastered—can elucidate the critical components of these skills, thus favoring the motor development process of children and young people.

Despite the model of Stodden et al. [11] to argue to motor competence (general FMS proficiency) and perception of motor competence in terms of state (how one is instead of how one improves), empirical evidence in adults [34,35] and models [36] guarantee space for improvement as a potential motivator in skill acquisition. It is true, nonetheless, that low self-perception, related to the current low skill level state, may decrease one’s motivation to even participate in activities, eliminating the potential to learn the skills or perceive the possibility for improvement. Thus, the potential psychological impact of varying competency levels of motor skills has potential ramifications not only for further skill development, but also for future physical activity participation.

Regardless of the current and previous [13,14] support to the idea of a proficiency barrier, some issues still need to be considered in light of the current literature. Despite not having classified the participants by FMS performance level or having investigated the proficiency barrier per se, the study of O’Keeffe et al. [28] speaks directly to the topic. A group of teenagers practiced the fundamental movement pattern of throwing for 3 weeks (180 min) while others practiced either a sport-specific skill (badminton overhead clear) or did not have specific practice (control group). The participants who practiced the FMS improved their performance in the practiced skill and also in two other specific sport skills: the javelin throw and the badminton overhead clear. In contrast, the group who practiced the sport-specific skill improved only the practiced skill, with no improvements being observed in the javelin throw or FMS throwing. These results seem to support the idea of a proficiency barrier as those who improved the FMS would have the basis or would have mastered the critical components to improve in the sport-specific skills. Yet, the fact that specialized skills did not necessarily transfer to enhanced FMS is an interesting finding. Despite such asymmetrical transfer being expected (improving FMS transfer to more complex skills while practicing more complex skills might not transfer to FMS), all groups in that experiment were stated to be at a “not-mature” or “near-mature” levels (the distribution indicates that one participant only was at this second level). Thus, the improvement in the badminton overhead clear, according to the proficiency barrier, should not occur.

Clearly, we should be cautious to reject the proficiency barrier based on these results. The proficiency levels were arbitrarily categorized and, thus, even the “not-mature throwers” could have mastered the critical components that would allow them to improve the sport-specific skills (the actual distribution was not provided in the paper). Another possibility is that our idea of a strict proficiency barrier might be wrong. Seefeldt [4] postulated that “children deprived of learning the fundamental skills have difficulty when they attempt to learn the transitional motor skills” (p. 316). We have assumed that children deprived of learning the FMS would not be able to learn the TMS. Pacheco et al. [14] and the current findings would support our view, but it might be that this only held in this sample. Nonetheless, one should attempt to replicate the mentioned study as to solve this potential counter-evidence to the proficiency barrier.

It is also interesting that in the O’Keefe et al. [28] study, practice in the sport-specific skills was not necessary for the FMS group. In this case, there was a direct influence from practicing FMS on TMS (transfer of practice). The transfer provides support to the overall shift in distribution of the FMS/TMS relation as demonstrated in Pacheco et al. [14]. That is, when someone learned the required components of the FMS, there is a degree of transfer to the more complex skills—the learner does not start from “zero”. Despite not having performed the same procedure as O’Keefe et al. [28], the results found by Costa et al. [13] allow us to say that children with better performance in throwing and volley did not need to learn from “zero” and, therefore, obtained better results in volleyball serve performance.

A limitation in the present manuscript should be mentioned. One might consider that using the same participants (and assessments) as Pacheco et al. (in press) would bias the data in favor of the proficiency barrier. However, the present analyses—which encompassed changes in performance after intervention—had a strong potential to modify the relationship found in the previous study as all children could directly improve the TMS. Therefore, if any future time point departed from the relationship observed here, then we would have sufficient evidence against the proficiency barrier and to state that Pacheco et al.’s (in press) results were spurious. 

In summary, our results provide evidence for the proficiency barrier proposition and, thus, provide evidence for a hierarchical nature of motor development. However, beyond the demonstration of a given developmental relation, our results also strengthen the argument for coaches and physical education teachers to promote FMS in their classes. Furthermore, identifying and teaching the critical antecedents of given skills can be a way to facilitate intra- and inter-task development. As improvement in the FMS (and thus the critical antecedents) can improve perception of competence [11,37], facilitating motivation to participate in different activities can enhance continued learning of FMS as well as promote physical activity. Along this line, future studies should focus on identifying critical antecedents of other FMS. 

While this study and Costa et al. [13] provide initial empirical evidence for the proficiency barrier, more work is needed to better understand the potential impact of other factors that may influence the ability to overcome a proficiency barrier (e.g., age, motivation, self-perceptions, sociocultural factors, etc.). In addition, longitudinal studies are needed to understand the potential broader impact of the proficiency barrier on other critical developmental outcomes [38,39]. 

## 5. Conclusions

The results of the present study are the first to provide evidence supporting the proficiency barrier hypothesis proposed by Seefeldt [4] encompassing all aspects of the proposition and with a longitudinal design. Our findings also alert coaches and teachers to the importance of developing the FMS critical antecedents before teaching the TMS or the sports skills. Interventions directed to mastering the critical antecedents might be the pathway needed to surpass the proficiency barrier, facilitating later improvements in TMS or specific sport skills performance.

## Figures and Tables

**Figure 1 ijerph-19-07184-f001:**
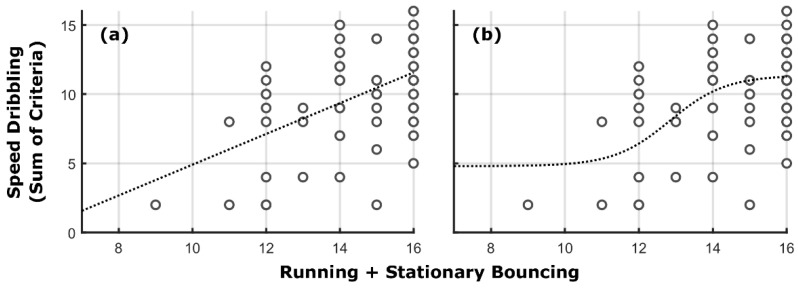
Relation between the sum of stationary dribbling and running components and speed dribbling components adjusted by (**a**) linear and (**b**) logistic functions. The logistic function was restricted to have $FMS_{b}=12.84$ and δ=0.7586. Each dot represents a single participant or an overlap of participants with the same performance.

**Figure 2 ijerph-19-07184-f002:**
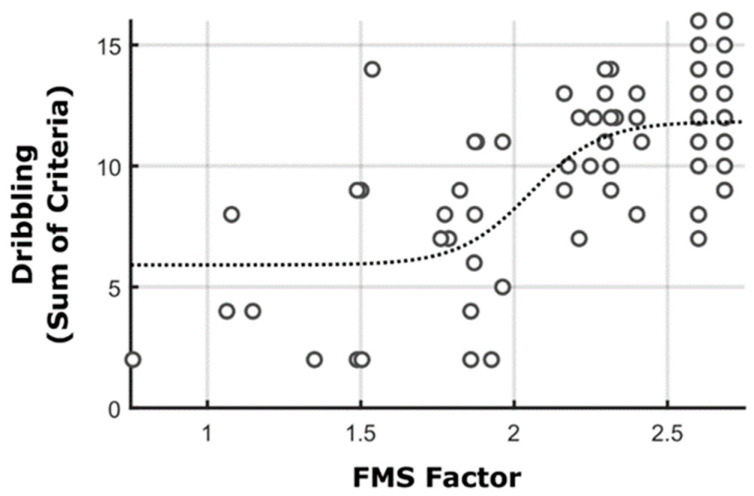
Relationship between the eigenvalues of the FMS factor found from Horn’s Parallel Analysis (considering the FMS data from the first data collection) and the speed dribbling performance in the second data collection. Each dot represents a single participant or an overlap of participants with the same performance.

**Table 1 ijerph-19-07184-t001:** Descriptive (mean ± 95% confidence interval) and inferential statistics (*t*-test) of running, leaping, hopping, stationary dribbling, catching, kicking, and speed dribbling in the first and second data collection.

Scheme 84	Baseline	Post-Intervention	*t*-Test (84)	*p*-Value
Running	7.59 ± 0.18	7.87 ± 0.10	3.63	<0.001
Leaping	4.12 ± 0.19	4.16 ± 0.19	0.41	0.523
Hopping	4.24 ± 0.29	4.21 ± 0.33	0.14	0.797
Stationary Dribbling	6.12 ± 0.41	6.84 ± 0.32	4.48	<0.001
Catching	4.56 ± 0.27	4.55 ± 0.26	0.08	0.928
Kicking	6.88 ± 0.28	7.14 ± 0.26	1.78	0.007
Speed Dribbling	8.91 ± 0.84	10.13 ± 0.75	4.69	<0.001

**Table 2 ijerph-19-07184-t002:** Cross-table between FMS (running and stationary dribbling) and TMS (speed dribbling) categories considering the first and second data collections for FMS.

		FMS Baseline		FMS 2nd Post Intervention	
	Criteria	≤11	>11	≤11	>11
TMS 2nd Data Collection	>9	0	54	0	54
	≤9	11	20	3	28

## Data Availability

The data is a property of the School of Physical Education and Sport, University of S. Paulo, Brazil, as is therefore protected from being freely shared. Yet, all researchers wanting to use the data will have to comply with the School policies. Further, they can address their request to the main author—Fernando Garbeloto dos Santos.

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
