# Peer review of "Testing Seefeldt’s Proficiency Barrier: A Longitudinal Study"

_ijerph, 2022, doi:10.3390/ijerph19127184_

Round 1

Reviewer 1 Report

This article has high novelty and has a good contribution value. In addition, the article still needs to be revised in several places:

The abstract does not account for whether the sample size was for normal children.

Line 110, with 10 consecutive teaching classes, once a week, the intervention took place between March and April of 2016. I think a weekly intervention is eventually needed for 2 1/2 months.

Line 116, Is it one physical education teacher for grades 2-5 or multiple? If multiple teachers are intervening, how do the authors consider individual differences among teachers, instructional styles that need to be considered, and differences in teachers' knowledge training from researchers?

Line 138, Here it is written that are assessed in children from 3 to 10 years of age, but above it says children 7-10 years old.

Reviewer 3 Report

Overall a very interesting study that develops the research topic and builds upon the knowledge within the field. 

Within the discussion opening paragraph (lines 270 - 275) you refer to 'hypothesis' and this was the study's aim. I believe this could have been presented in more detail within your aims of research study (lines 85 - 92) to ensure the readers clarity. 

Within your methodology:

It would be good to see the location of this study population described (i.e UK, USA) and an understanding of their prior PE experience (if possible, for example has it been compulsory for the participants to engage in 1 hour of PE per week since beginning school) 

Is there a control group or school used as comparison or was it simply pre and post data that has been used to determine success of intervention

Line 105 states (~30% of the total 105 number of children) – what is this in reference too

Reviewer 4 Report

Excellent manuscript that brings great value to the field for application and direction.  I listed below the elements of grammar that needed to be adjusted:

 Line 13: there “is” no study or there are no studies

Line 17: to 10 years participated “in” the study.

Line 23: the barrier can be captured through
